# Genomic Characterization of Multidrug-Resistant Pathogenic Enteric Bacteria from Healthy Children in Osun State, Nigeria

**DOI:** 10.3390/microorganisms12030505

**Published:** 2024-03-01

**Authors:** Jessica N. Uwanibe, Idowu B. Olawoye, Christian T. Happi, Onikepe A. Folarin

**Affiliations:** 1African Center of Excellence for Genomics of Infectious Diseases (ACEGID), Redeemer’s University, Oshogbo 232102, Osun State, Nigeria; uwanibej@run.edu.ng (J.N.U.); olawoyei0303@run.edu.ng (I.B.O.); happic@run.edu.ng (C.T.H.); 2Department of Biological Sciences, College of Natural Sciences, Redeemer’s University, Oshogbo 232102, Osun State, Nigeria

**Keywords:** whole genome sequencing, multidrug resistance, AMR genes, plasmid, virulence, enterobacteriaceae, healthy children

## Abstract

Antimicrobial resistance (AMR) is responsible for the spread and persistence of bacterial infections. Surveillance of AMR in healthy individuals is usually not considered, though these individuals serve as reservoirs for continuous disease transmission. Therefore, it is essential to conduct epidemiological surveillance of AMR in healthy individuals to fully understand the dynamics of AMR transmission in Nigeria. Thirteen multidrug-resistant *Citrobacter* spp., *Enterobacter* spp., *Klebsiella pneumoniae*, and *Escherichia coli* isolated from stool samples of healthy children were subjected to whole genome sequencing (WGS) using Illumina and Oxford nanopore sequencing platforms. A bioinformatics analysis revealed antimicrobial resistance genes such as the *pmrB_Y358N* gene responsible for colistin resistance detected in *E. coli* ST219, virulence genes such as *senB*, and *ybtP&Q*, and plasmids in the isolates sequenced. All isolates harbored more than three plasmid replicons of either the Col and/or Inc type. Plasmid reconstruction revealed an integrated *tetA* gene, a toxin production *caa* gene in two *E. coli* isolates, and a *cusC* gene in *K. quasivariicola* ST3879, which induces neonatal meningitis. The global spread of AMR pathogenic enteric bacteria is of concern, and surveillance should be extended to healthy individuals, especially children. WGS for epidemiological surveillance will improve the detection of AMR pathogens for management and control.

## 1. Introduction

Enterobacteriaceae are inhabitants of the gastrointestinal tract, sometimes called enteric bacteria. Most are commensals, such as gut flora or microbiota, and some are known to be pathogenic, mainly when found in some specific areas of the human body [1]. Enteric pathogens significantly cause morbidity and mortality among young children, especially in low and middle-income countries (LMICs), as well as in adults. However, many enteric bacterial infections are either asymptomatic or result in only mild-to-moderate disease [2]. Some common enteric pathogens found in children are *Citrobacter freundii*, *Klebsiella* spp., *Escherichia coli*, *Enterobacter* spp., *Salmonella typhi*, and *Shigella* spp. [3,4].

The exact burden of diseases caused by enteric bacteria in Sub-Saharan Africa is vague due to poor healthcare services and lack of proper data documentation, to mention a few reasons. Recent reports have shown an increase in the prevalence of diseases caused by enteric bacteria, such as sepsis [5], typhoid [6], cholera [7], shigellosis [8], and diarrheal infection [9]. This is of great concern, especially regarding proposed disease elimination and meeting the third sustainable development goal (SDG3), good health and well-being. It is, therefore, of great importance to embark on active surveillance of these pathogens for management and control if this goal is to be met by 2030.

Antimicrobial resistance (AMR) has been established to be a significant driver for the persistence and spread of enteric diseases, particularly in cases where the causative agents are multidrug-resistant (MDR) or extensively drug-resistant (XDR) pathogens [10,11] as MDR pathogens are known to be non-susceptible to one or more drugs in three antibiotics classes, while XDR pathogens are non-susceptible to at least one drug in all but two or fewer antibiotics classes. The overuse and misuse of antibiotics have been identified as significant causes of AMR in enteric bacteria, as these create selective pressure and promote the emergence of resistance, leading to an alarming increase in the prevalence of MDR enteric bacteria among healthy individuals [12]. In addition, the ease of transfer of plasmids and transposons harboring genes associated with AMR amongst enteric bacteria through horizontal gene transfer enhances the persistence and spread of AMR [13,14].

Currently, most management and control efforts on infectious diseases are focused on symptomatic cases. It is essential to equally consider and investigate asymptomatic individuals, especially in LMIC, where enteric diseases are endemic and transmission is high due to poor hygiene. Furthermore, asymptomatic carriers of pathogenic enteric bacteria are rarely focused on in research studies, even though they are crucial in disease transmission and persistence, serving as reservoirs for antimicrobial-resistant genes [15,16]. A healthy human microbiome is one of the major reservoirs of antibiotic-resistant genes (ARGs) which are transmissible to other pathogenic bacteria [14,17]. Recent research reveals that ARGs contribute to the prevalence of MDR enteric bacteria in healthy humans, ensuring the continuous circulation of ARGs in the population [18]. Children constitute a significant population burdened by MDR enteric infections [19,20,21,22]. In addition, reports have shown AMR enteric bacteria to be a significant cause of mortality in neonatal sepsis in Nigeria, despite government interventions and policies [22,23,24,25]. Furthermore, the country has only a few in-depth genomic analyses of enteric bacteria that are of public health interest [26,27,28].

Genomic surveillance of AMR enteric bacteria that are of public health importance, such as *Escherichia coli*, *Klebsiella pneumoniae*, *Salmonella typhi*, *Vibrio cholera*, and *Shigella* spp., not only informs treatment guidelines for diseases caused by these human pathogens but is necessary for the design and implementation of AMR interventions and control measures [29]. Whole genome sequencing (WGS) provides a high-resolution method capable of providing in-depth genomic characterization and epidemiological information about a pathogen. This further sheds more light on genomic diversity, transmission dynamics, evolution, AMR dynamics, and spread, aiding in better control measures [30]. This has been implemented in studies of asymptomatic carriers to characterize the pathogens further, understand the actual dynamics of AMR, and investigate continuous pathogen development within the natural host [14,31,32].

Thus, this study utilized a whole genome sequencing (WGS) approach to characterize AMR genes and mobile genetic elements in isolates obtained from fecal samples to understand the dynamics of AMR enteric pathogens in healthy children from Osun State, Nigeria.

## 2. Methods

### 2.1. Ethical Approval

Ethical clearance for this study was obtained from the research ethics committee of the Ladoke Akintola University of Technology Teaching Hospital (LAUTECH) (LTH/EC/2019/09/431) and the Ministry of Health (OSHREC/PRS/569T/164), Osogbo, Osun State, Nigeria.

### 2.2. Bacteria Isolation, Identification, and Selection

A total of 147 fecal samples were collected from presumptively healthy children between 1 and 15 years of age between 2019 and 2020 in a study focused on typhoid fever’s seroprevalence and the genomic characterization of enteric pathogens in healthy children. Serological data for typhoid fever prevalence have been previously published [33]. Fecal samples were cultured on Salmonella/Shigella and MacConkey agar. Initial identification was performed with an API 20E test kit (BioMeriuxe, Marcy-l’Étoile, France) for 52 positive enteric cultures. Thirteen (13) enteric isolates were selected based on their antibiotic resistance profile for genomic characterization.

### 2.3. Antimicrobial Susceptibility Test of the Isolates

Antimicrobial susceptibility testing was performed using the disk diffusion technique. The antibiotics disc used were Gentamicin (30 µg), Tetracycline (30 µg), Ciprofloxacin (5 µg), Chloramphenicol (30 µg), Trimethoprim/sulfamethoxazole (25 µg), Ceftazidime (30 µg), Ceftriaxone (30 µg), Cefotaxime (30 µg), and Aztreonam (30 µg) as described previously by [34]. Results were interpreted according to the 2017 guidelines provided by the Clinical and Laboratory Standards Institute (CLSI). Results were analyzed and interpreted using the ABIS online software v12 (http://www.tgw1916.net/bacteria_logare_desktop.html) accessed on 22 May 2021 [35].

### 2.4. Whole Genome Sequencing

The isolates were subcultured before DNA extraction. DNA was extracted using a Qiagen DNeasy Blood and Tissue kit (Qiagen, Hilden, Germany). Extracted samples were quantified using a dsDNA high-sensitivity assay kit on a Qubit fluorometer (ThermoFisher Scientific, Waltham MA, USA)). Sequencing libraries were prepared using a Nextera DNA flex preparation kit (Illumina, San Diego, CA, USA). Library preparation protocol was adopted from the CDC PulseNet Nextera DNA Flex Standard operating protocol and sequenced using a Illumina Miseq platform and NextSeq 1000/2000 at the African Center of Excellence for Genomics of Infectious Diseases (ACEGID), Redeemer’s University, Nigeria.

To improve the plasmid assembly, we performed a single run on a GridION x5 to generate long reads. Library preparation and sequencing were performed using a Rapid PCR Barcoding kit (SQK-RPB004) (Oxford Nanopore Technologies, Oxford, UK), following the manufacturer’s recommendations. We used a GridION MK1 sequencer, FLO-MIN10 6D R9 flow cell, and MinKNOW software v22.10.7 for sequencing.

### 2.5. Genomic Data Analysis

Raw FASTQ files were processed with the Connecticut Public Health Laboratory (CT-PHL) pipeline, also known as C-BIRD v0.9 “https://github.com/Kincekara/C-BIRD (accessed on 9 May 2023)”. The assembled contigs were further checked for contamination using CheckM [36]. Isolates with a bracken taxon ratio <0.7, genome estimated ratio >1.1, estimated sequencing depth <40×, and genome completeness <90% were excluded from subsequent analyses, as those isolates were either deemed contaminated with other bacterial species, had an unusual larger genome size, had low genome coverage, or had a low sequencing depth. Further analyses, including genome annotation, plasmid detection, antimicrobial resistance, virulence prediction, and MLST typing, were performed with the Public Health Bacterial Genomics (PHBG) v1.3.0 “https://github.com/theiagen/public_health_bacterial_genomics (accessed on 9 May 2023)”. Genome assemblies of isolates were refined with a unicycler v0.4.9 [37] hybrid assembler using Illumina and Oxford nanopore sequence reads. The plasmids were reconstructed and typed with MOB-suite v3.1.0 [38] and annotated with pLannotate v1.2.0 [39] and Prokka v1.14.6 [40].

## 3. Result

### 3.1. Isolate of Bacteria Strains

The selected 13 multidrug-resistant isolates were *Enterobacter hormaechei* (*n* = 1), *Citrobacter sp. FDAARGOS_156* (*n* = 1), *Enterobacter cloacae* (*n* = 1), *Klebsiella quasivariicola* (*n* = 1), *Klebsiella pneumoniae* (*n* = 3), and *Escherichia coli* (*n* = 6). All were isolated from fecal samples obtained from healthy children.

### 3.2. Antibiotic Susceptibility Test and AMR Prediction

The highest resistance (85%) was observed in ciprofloxacin and cefotaxime, and the least resistance (46%) was seen in tetracycline. All 13 isolates subjected to antibiotic susceptibility tests were multidrug-resistant, defined as resistance to two or more classes of antibiotics (Table 1).

### 3.3. Genome Sequence Analysis of the Isolates

Genome sequencing data based on the taxon ratio analysis showed all isolates harbored a wide range of AMR genes, plasmid replicons, and virulent genes. All of the isolates harbored beta-lactam-resistant genes, with 23% of the isolates seen to harbor tetracycline-resistant genes *tetA*, and macrolides pmrB_R256G and pmrB_Y358N resistance genes, which are responsible for colistin resistance via the efflux pump mechanism. Also, 92% of the isolates had different efflux pump genes responsible for at least two classes of antibiotic resistance (Table 2). Virulence genes were seen in 62% of the isolates, while plasmids were detected in 92% of the isolates detailed in Table 2.

We then considered sequence data based on the genome estimated ratio > 1.1, estimated sequencing depth <40×, and genome completeness (<90%). Only five out of the thirteen sequenced isolates passed this quality control step. These isolates asterisked in Table 2 (*E. coli* = 3, *K. quasiveriicola* = 1, and *E. hormaechei* = 1) were selected for further downstream sequence analysis and characterization. The sequence data have been submitted to NCBI under the BioProject accession number PRJNA838568.

Genome lengths of 4.8 Mbp to 6 Mbp, 5.6 Mbp, and 5.8 Mbp for *E. coli*, *K. quasiveriicola*, and *E. hormaechei* were observed. The assembled contigs of the bacteria genome antimicrobial resistance (AMR) detection software (AMRFinder+ v3.10.42 and Kleborate v2.0.4) embedded in the PHBG pipeline showed the presence of beta-lactam-resistance genes such as *bla*TEM-1, *bla*EC, *bla*ACT, and *bla*OKP-D, (Table 1). In addition, all five isolates contained other AMR genes associated with either fluoroquinolone (*gyrA*S83L, *qnrS1*), aminoglycoside (*aph(6)-Id*, *aph(3*″*)-Ib*), tetracycline (*tetA*), or sulfonamide (*sul1*, *sul2*) resistance. Multilocus sequence typing (MLST) predicted the ST type for three isolates, *E. coli* ST 219, ST450, and *K. quasiveriicola* ST 3897, while the remaining two isolates, *E. hormaechei*, and *E. coli*, had no ST prediction. This study revealed the presence of fluoroquinolone-resistant genes with more than triple mutations (*gyrA_*S83L, *gyrA_*D87N, *parC_*S80I, and *parE_*S458A) in *E. coli* ST219 and the Colistin-resistant gene (*pmrB_*Y358N) in *E. coli* ST450 reported in XDR *E. coli.*

### 3.4. Detection and Characterization of Virulence Genes and Mobile Genetic Elements in the Five Isolates

Of the five whole genome sequences that passed the quality control step, virulence genes were detected in all three *E. coli* isolates. In contrast, none were detected in *K. quasivariicola* and *E. hormaechei* (Table 2). Virulence genes responsible for bacteria iron uptake were detected in all the *E. coli* isolates. Aerobactin virulence genes *iucABCD* and *iutA* were also present in all the *E. coli* isolates, but only two *E. coli* isolates harbored the Yersiniabactin virulence genes *ybtP* and *ybtQ* (Table 2). Other virulence genes detected include the gene-encoding enterotoxin *senB*, the increased serum survival *iss* gene *sigA*, the secreted autotransporter toxin -*sat*, the east-1 heat-stable toxin -*astA*, the long polar fimbria -*ipfA*, the *LEE* encoded type III secretion system effector- *espX1*, for adherence- *fdeC*, the P-fimbria operon- *papHCFGII*, the Salmonella HilA homolog -*eilA*, and the iron-regulated outer membrane virulence protein–*ireA* gene.

A total of 28 plasmids were also detected in these five isolates, with each harboring between three and nine plasmid replicons known to be associated with antibiotic-resistant genes (ARGs). The Inc plasmid type is the most common plasmid type across all isolates. IncF and Col plasmids were detected in 90% (4/5) of the isolates. The IncF replicons detected included IncFII(pBK30683) (1/5), InFIB(K) (2/5), IncFII(K) (1/5), IncFII(pECLA) (1/5), IncFIA(HI1) (1/5), IncFIA (1/5), IncFIB(AP001918) (1/5), and IncFII(pRSB107) (1/5). The Col plasmid detected had replicons which included Col440I (2/5), Col440II (1/5), ColpHAD28 (3/5), ColMG828 (1/5), and Col156 (1/5). Other plasmid replicons detected were IncR (1/5), IncI2 (1/5), IncQ1 (1/5), and IncB/O/K/Z (2/5).

### 3.5. Plasmid Reconstruction Using Oxford Nanopore Sequencing Platform

We reconstructed 13 plasmids from 3536 to 163,036 bp detected in *E. hormaechei*, *K. quasivariicola*, and two *E. coli* isolates (Appendix A). Plasmid reconstruction for the other *E. coli* was not conducted due to low sequencing library concentration. The *K. quasivariicola* plasmids were either conjugative, non-mobilizable, or mobilizable. The largest plasmid contained 11 contigs with IncFIB, IncFII, rep_cluster_2183, rep_cluster_2327, and rep_cluster_2358 replicon types. None of the plasmids in *K. quasivariicola* harbored any antibiotic-resistance genes; however, several metal-resistance genes and efflux proteins such as *copA*, *copB*, *silP*, *cusA*, *cusB*, and *cusC* were integrated into the plasmid (Figure 1). Interestingly, we detected the tetracycline resistance gene *tetA* and some mercury resistance genes such as *merA* and *merC* integrated into the plasmids of the two *E. coli* isolates that code for colicin polypeptide toxins, while the *E. hormaechei* isolate had plasmids carrying *ylpA* virulence genes (Figure 1 and Appendix A).

## 4. Discussion

Antimicrobial resistance, especially in enteric bacteria that are of public health interest, continues to be of global concern, especially in low and middle-income countries [35]. In Nigeria, there is a paucity of information on the genomic characterization of circulating multidrug-resistant enteric bacteria, especially in healthy individuals that serve as reservoirs. Therefore, we recovered enteric pathogens that are of public health interest in this study from presumptively healthy children such as *Enterobacter hormaechei*, *Citrobacter* sp. *FDAARGOS_156*, *Enterobacter cloacae*, *Klebsiella quasivariicola*, *Klebsiella pneumoniae*, and *Escherichia coli.* Most importantly, all the isolates harbored antibiotic-resistant genes for at least one class of antibiotics; some harbored virulence genes and plasmid replicons carrying resistant and virulence genes. The presence of these pathogens and, importantly, the carriage of AMR genes, plasmids, and virulence genes conferring pathogenicity and resistant phenotypes in these healthy children are of great concern to public health. The occurrence of these pathogenic enteric bacteria in febrile individuals is well documented across the country [22,35,41,42]; however, there is limited information on healthy individuals in Sub-Saharan Africa as far as the authors know, which makes this study one of the very few studies with a genomic surveillance report on healthy children in Nigeria.

Similar resistant genes (*gyrA_*S83L, *gyrA_*D87N, *parC_*S80I, and *parE_*S458A) identified from this study in presumptively healthy children have been reported in disease cases of children admitted to hospitals in Bangladesh and Benin [43,44]. The presence of these enteric pathogens isolated from presumptive healthy children is of great concern as these children serve as potential reservoirs for transmission of these pathogens to other susceptible individuals. In addition to transmission, the presence of resistance genes indicates the possibility of treatment failure, further promoting the spread of AMR in society and its longer existence in the population. It is possible that these pathogens developed resistance due to consistent drug pressure, as most drugs are readily available to citizens in the country, encouraging indiscriminate use of these drugs [18].

Our study provides a glimpse into the genetic variation of enteric pathogens in healthy children in Osun State in the Southwestern part of Nigeria. Virulence genes identified amongst the *E. coli* isolates are known to significantly influence the degree of pathogenicity of bacterial infection in confirmed disease patients in some reported cases [45,46]. It is unclear how these genes in the bacteria are not causing any symptoms in these children. However, reports have shown that colonization with AMR enteric pathogens in healthy individuals could lead to pulmonary infection, urinary infection, and bacteremia [47,48]. *K. quasivariicola* and *E. hormaechei* harbored no virulence genes. The presence of more than three plasmid replicons associated with AMR is of significant concern as reports of colonizing enteric bacteria resulting in infections are facilitated by MDR and plasmid carriage [49].

The pathogenic enteric bacteria in this study are known as commensals of the gut microbiota and have been reported in other studies [22,50]. Once seeded with AMR, commensal organisms may significantly contribute to the dissemination of resistance due to the connectivity in microbial communities [51]. The enteric bacteria (*E. coli*, *K. quasiveriicola*, and *E. hormaechei*) detected in this study have been reported in many cases to cause diarrhea and sepsis in children and also serve as colonizers in healthy children [4,52,53]. In addition, the pathogens were observed to be resistant to almost all of the classes of antibiotics. The presence of AMR genes and plasmids harbored by these bacteria in healthy children further confirms the in vitro resistance observed, and it is of concern as children in this community will enable the persistent transmission of diseases caused by these enteric bacteria, leading to increased community-acquired infections [54]. This could increase morbidity, mortality, and healthcare costs, especially in immunocompromised individuals. This, therefore, necessitates active surveillance of these AMR bacteria in both asymptomatic and clinical cases to give a total overview of the transmission dynamics and evolution.

Although not surprising, the detection of AMR genes in this population of healthy children is alarming, as antibiotic abuse has been established in this part of the world [26,55]. This is seen in the AST result and genomic data, as most isolates were resistant to folate-pathway drugs, quinolones, and beta-lactam drugs, typically first and second-line antibiotics. The presence of efflux pump genes responsible for MDR resistance increases the expression of resistance in these isolates. Resistance to these drugs, some of the most common antibiotics used in treating diarrhea diseases in children, could pose a challenge in future infections with these resistant strains. Also, reports have shown that most cases of infection with MDR enteric bacteria are not just from a person-to-person transmission or from contaminated water/food, but could also be transmitted from the individual colonized by these MDR bacteria through horizontal gene transfer [49]. This is worrisome as almost all isolates sequenced in this study, as seen in Table 1, harbored AMR genes for beta-lactams, aminoglycosides, quinolones, colistin, and fosfomycin drugs.

A significant driver of the spread of antibiotic resistance in bacteria is the presence of mobile genetic elements [56]. Plasmids spread ARG to other bacteria through horizontal gene transfer, increasing antibiotic resistance prevalence. They are known to contain genes responsible for antibiotic resistance, colonization, and virulence, which provides an advantage for bacteria survival [56]. As seen in this study, the plasmid recovered from the *E. coli* ST219 strains harbored a *tetA* gene responsible for tetracycline resistance and a *caa* gene that codes for colicin polypeptide toxins known to destroy cells of other organisms by depolarizing the cytoplasmic membrane of the cells [57,58]. Also, the plasmid in the *K. quasivariicola* ST3897 harbored the *cusC* gene associated with copper and silver resistance, which, when found in pathogenic *K. pneumoniae*, facilitates the invasion of the brain microvascular endothelial cells, thereby causing neonatal meningitis [59,60].

Apart from the presence of virulence genes and mobile genetic elements as a driver of MDR, the indiscriminate use of antibiotics in this part of the world is also a significant contributing factor to the high prevalence of MDR bacteria [27]. Although the healthy children in this study were not on any antibiotics drugs at the time of sample collection, it has been reported that children are exposed to antibiotics early on, mainly without a physician’s prescription [61,62,63]. To mitigate the continuous spread of MDR enteric bacteria and the possible emergence of XDR enteric bacteria, there is a need for proper surveillance in healthy individuals to achieve proper monitoring and control of AMR in the country. Although some control measures such as infection prevention control and antimicrobial stewardship have been implemented to monitor AMR in the population [55,64,65], the non-inclusion of healthy individuals in these control measures will truncate all efforts. Another critical factor in proper surveillance is the use of sensitive techniques. In addition to confirming the AST analysis, the whole genome sequencing employed in this study revealed genetic elements that drive pathogenicity and the spread of MDR enteric bacteria from healthy children. This might not have been possible by culture or a PCR alone. This proves the importance of the WGS technique in pathogen enteric bacteria surveillance and its implementation in healthy individuals to achieve disease elimination and eradication.

### Limitation

The limitation of this study is the small number of the isolates (13) considered for the whole genome sequence from the overall sample size or pure isolates obtained from the fecal samples. The molecular characterization detected provided an insight into the existence of AMR genes and other mobile elements that may confer pathogenicity in these organisms and can further be transmitted to other organisms. Therefore, this study serves as an indicator for the general and adequate national surveillance of enteric bacteria in healthy individuals, especially children, for the total control of AMR. We therefore recommend further studies on the AMR of healthy individuals, especially children who are the major population at risk of AMR infection, to ensure proper monitoring and management policies.

## 5. Conclusions

Our findings in this study showed the presence of MDR enteric bacteria harboring resistant genes, virulence genes, and plasmids in healthy children. This may contribute to the continuous, global, and widespread increase in AMR observed. Therefore, close inspection and surveillance of the healthy population in addition to clinical cases is recommended to control the spread and ultimately achieve eradication of disease. Applying WGS in epidemiological surveillance will improve the detection of MDR pathogens by overcoming the limitation of analyzing only a small part of the genome and providing more rapid management, thus controlling the emergence of new antibiotic-resistant strains and their evolution.

## Figures and Tables

**Figure 1 microorganisms-12-00505-f001:**
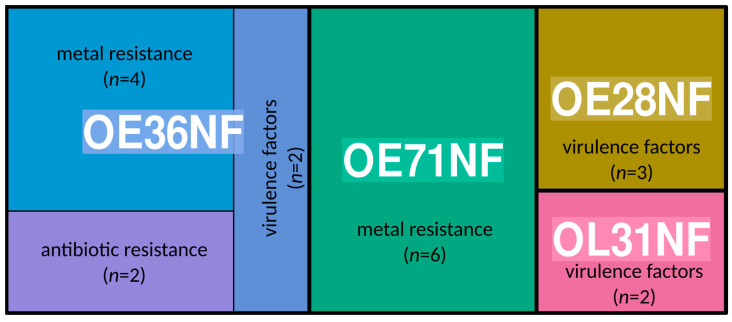
Treemap showing the proportion of virulence factors, metal resistance, and antibiotic resistance genes curated from plasmids (*n* = 13) of the isolates that were successfully sequenced on the Oxford Nanopore (OE36NF-*E. coli*, OE71NF-*K. quasipneumoniae*, OE28NF-*E. coli*, and OL31NF-*E. hormaechei*).

**Table 1 microorganisms-12-00505-t001:** Multiple antibiotic resistance (MAR) pattern of enteric isolates from healthy children in Osun State.

Sample ID	Species	Antibiotics Profile (Resistant)
OL31NF	*Enterobacter hormaechei*	ATM; CTX; CAZ; SXT; CIP.
OL13NFPC1	*Escherichia coli*	ATM; CRO; CTX; SXT; C.
OL19NF	*Citrobacter* sp. *FDAARGOS_156*	ATM; CRO; CTX; CAZ; SXT; C.
OL44NF	*Escherichia coli*	TE; SXT; CIP; CN.
OE28NF	*Escherichia coli*	ATM; CRO; CTX; TE; SXT; CIP; CN; C
OE36NF	*Escherichia coli*	CTX; TE; SXT; CIP; CN.
OE41NF	*Escherichia coli*	CTX; CIP; C.
OE43NF	*Enterobacter cloacae*	ATM; CRO; CTX; CAZ; TE; SXT; CIP; CN.
OE54NF	*Klebsiella pneumoniae*	ATM; CRO; CTX; CAZ; CIP; CN.
OE71NF	*Klebsiella quasivariicola*	CAZ; CIP; C; CN.
OE73NF	*Klebsiella pneumoniae*	ATM; CRO; CTX; CAZ; TET; SXT; CIP; C; CN.
OE75NF	*Klebsiella pneumoniae*	CTX; SXT; CIP; CN
J21	*Escherichia coli*	ATM; CRO; CTX; CAZ; TE; SXT; CIP; CN; C.

ATM: Aztreonam, CRO: Ceftriaxone, CTX: Cefotaxime, CAZ: Ceftazidime, TE: Tetracycline, SXT: Trimethoprim/sulfamethoxazole, CIP: Ciprofloxacin, CN: Gentamicin, C: Chloramphenicol.

**Table 2 microorganisms-12-00505-t002:** Summary of genomic characterization of whole-genome-sequenced Enteric isolates.

S/N	Sample ID	Species	Sequence Type	AMR Genes	Virulence Genes	Plasmid Replicons
1	OL31NF *	*Enterobacter hormaechei*	No ST predicted	blaACT-16, qnrS1, fosA, oqxB, oqxA	No VIRULENCE genes detected by NCBI-AMRFinderPlus	Col(pHAD28), IncFIB(K), IncFII(pECLA)
2	OL13NFPC1	*Escherichia coli*	ST409	acrF, mdtM, glpT_E448K, blaEC	espX1	No plasmids detected in database
3	OL19NF	*Citrobacter* sp. *FDAARGOS_156*	ST187	blaCMY	No VIRULENCE genes detected by NCBI-AMRFinderPlus	Col(pHAD28), ColRNAI
4	OL44NF	*Escherichia coli*	No ST predicted	blaEC, aadA1, dfrA7, sul1, tet(A), aph(3′)-Ia, mdtM, pmrB_Y358N, acrF, qnrS, blaTEM-1, aph(6)-Id, aph(3″)-Ib, sul2, qnrB19, mdtM	nfaE, afaC, lpfA, senB, iss, espX1, papA, sat, iutA, iucD, iucC, iucB, iucA, iha, f17a, f17g, ybtQ, ybtP, lpfA	Col(MG828), Col(MP18), Col(pHAD28), Col(pHAD28), Col(pHAD28), Col(pHAD28), Col156, IncFIB(AP001918), IncFII(pRSB107), IncI(Gamma), IncQ1
5	OE28NF *	*Escherichia coli*	No ST predicted	mdtM, blaEC, blaEC, blaTEM, acrF, tet(A)	espX1, iutA, iucD, iucC, iucB, iucA, fdeC, lpfA, iss, ireA, sigA, iha, lpfA, ybtP, ybtQ	Col(pHAD28), Col(pHAD28), IncB/O/K/Z, IncFIA(HI1), IncR
6	OE36NF *	*Escherichia coli*	ST 219	blaTEM-1, pmrB_Y358N, mdtM, acrF, emrD, gyrA_S83L, glpT_E448K, blaEC, sul2, aph(3″)-Ib, aph(6)-Id, sul1, dfrA7, tet(A)	eilA, lpfA1, iucA, iucB, iucC, iucD, iutA, iha, sigA, espX1, fdeC	IncB/O/K/Z, IncI2, IncQ1
7	OE41NF	*Escherichia coli*	No ST predicted	qnrB, oqxA, emrD, fosA7, glpT_E448K, mdtM, blaCMY, blaEC	iss, espX1, fdeC	Col156, IncFIA(HI1), IncFIB(AP001918), IncFIB(K), IncFIB(pHCM2), IncY
8	OE43NF	*Enterobacter cloacae*	ST1236	emrD, satA, mdtM, blaCMH, vmlR, aadK, mphK, blaEC, blaCTX-M	lpfA1, espX1, eilA	IncFII(K), IncR
9	OE54NF	*Klebsiella pneumoniae*	No ST predicted	fosA, oqxB, oqxA, emrD, blaSHV	No VIRULENCE genes detected by NCBI-AMRFinderPlus	Col440II, IncFIA(HI1), IncFIB(K)
10	OE71NF *	*Klebsiella quasivariicola*	ST3897	emrD, fosA, kdeA, oqxB, oqxA, blaOKP-D	No VIRULENCE genes detected by NCBI-AMRFinderPlus	Col440I, Col440II, FII(pBK30683), IncFIB(K), IncFII(K)
11	OE73NF	*Klebsiella pneumoniae*	ST185	blaTEM, fosA, blaSHV-11, dfrA15, aadA1, sul1, oqxB, oqxA, emrD	alo, plcR	Col(pHAD28), Col440I, IncFIB(K), IncFII(pKP91), IncR
12	OE75NF	*Klebsiella pneumoniae*	No ST predicted	fosA, emrD, pmrB_R256G, blaOKP-B, blaSHV-11, oqxB, oqxA, fosA	No VIRULENCE genes detected by NCBI-AMRFinderPlus	IncFIA(HI1), IncFIB(K), IncFIB(pNDM-Mar), IncFII(K), IncHI1B(pNDM-MAR)
13	J21 *	*Escherichia coli*	ST450	blaTEM-1, parC_S80I, parE_S458A, catA1, gyrA_D87N, gyrA_S83L, dfrA17, acrF, mdtM, blaEC, aadA1, blaOXA-1, qepA4, tet(A), aph(6)-Id, aph(3″)-Ib, sul2	sigA, sat, astA, ybtQ, ybtP, espX1, fdeC, iha, papH, papC, papF, papG-II, iucA, iucB, iucC, iucD, iutA, senB	Col(MG828), Col156, IncFIA, IncFIB(AP001918), IncFII(pRSB107), IncFII(pRSB107)

* Samples which passed the QC step.

## Data Availability

Sequence data of isolates are deposited in the NCBI Sequence Read Archive (SRA) under BioProject accession number PRJNA838568.

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
