# Peer review of "Genomic Characterization of Multidrug-Resistant Pathogenic Enteric Bacteria from Healthy Children in Osun State, Nigeria"

_microorganisms, 2024, doi:10.3390/microorganisms12030505_

Round 1

Reviewer 1 Report

Comments and Suggestions for Authors

1. The objective of this study was formulated: “Thus, this study utilized a whole genome sequencing (WGS) approach to characterize AMR genes and mobile genetic elements in isolates obtained from fecal samples to understand the dynamics of AMR enteric pathogens in healthy children from Osun state, Nigeria.

The study collection is small (only 13 isolates furthermore from 4 different species). In this sense the results are of interest but the sample may non-representative on the whole, and even less if you wish to study dynamics. The objective should be revised to sound more realistically, to correspond to study design.

2. The topic is not particularly novel, e.g. see these papers, some included children

https://pubmed.ncbi.nlm.nih.gov/34374428/

https://pubmed.ncbi.nlm.nih.gov/30226162/

https://pubmed.ncbi.nlm.nih.gov/31176748/

3. Limitations section is absent. Limitation is a small sample size and not clear how these results are representative of the situation in this country.

Author Response

  1. The objective of this study was formulated: “Thus, this study utilized a whole genome sequencing (WGS) approach to characterize AMR genes and mobile genetic elements in isolates obtained from fecal samples to understand the dynamics of AMR enteric pathogens in healthy children from Osun state, Nigeria.”

The study collection is small (only 13 isolates furthermore from 4 different species). In this sense the results are of interest but the sample may non-representative on the whole, and even less if you wish to study dynamics. The objective should be revised to sound more realistically, to correspond to study design.

Response: The study population was from a total of 147 healthy asymptomatic children, of which we had 57 different isolates that were subjected to AST. The isolates selected for sequencing were based on the AST outcome. Authors agree that the sample size is small for a generalized results but we believe that the results provided from the 13 isolates indicate a glimpse/insight  of the genomic profile of Enteric bacteria in healthy children living in the state. The statement of the objective now reads “Thus, this study utilized a whole genome sequencing (WGS) approach to characterize AMR genes and mobile genetic elements in isolates obtained from fecal samples to provide insight and understand the dynamics of AMR enteric pathogens in healthy children from Osun state, Nigeria.” (Lines 81-84).

  1. The topic is not particularly novel, e.g. see these papers, some included children

https://pubmed.ncbi.nlm.nih.gov/34374428/

https://pubmed.ncbi.nlm.nih.gov/30226162/

https://pubmed.ncbi.nlm.nih.gov/31176748/

Response: We agree with the Reviewer that the study is not novel globally as shared in the references provided. However, information on the AMR of Enterobacteriaceae in healthy children in Sub-Saharan Africa is very limited, and this study provides insight into the situation in children from Osun state, Nigeria. Lines 257 - 259 have been rephrased to read: “However, there is limited information on healthy individuals in Sub-saharan Africa as far as the authors know, which makes this study one of the very few studies with genomic surveillance report on healthy children in Nigeria.”

  1. Limitations section is absent. Limitation is a small sample size and not clear how these results are representative of the situation in this country.

Response: We apologize for the oversight; we have now included a paragraph to highlight the limitations of our study in the discussion section. The statement on limitation reads thus

 “Limitation

The limitation of this study is the small number of the isolates (13) considered for whole genome sequence from the overall sample size or pure isolates obtained in the fecal samples. The molecular characterisation detected provided an insight of existence of AMR genes and other mobile elements that may confer pathogenicity in these organisms and can further be transmitted to other organisms. Therefore, this study serves as an indicator for general and adequate national surveillance of enterics in healthy individuals especially children for the total control of AMR. We therefore recommend further studies on the AMR in healthy individuals especially children who are the major population at risk of AMR infection for proper monitoring and management policy” as shown on line Lines 345 – 354.

Reviewer 2 Report

Comments and Suggestions for Authors

The research addressed the main question that the WGS led characterize AMR genes and mobile genetic elements in isolates obtained from fecal samples. The genomic surveillance of AMR enteric bacteria with impact in health public for the design and implementation of AMR interventions and control measures are original and relevant for the field.  Antimicrobial resistance genes are responsible for colistin resistance in E. coli ST219, virulence genes, and plasmid replicons of the Col.  cusC gene in K. quasivariicola ST3879, which induces neonatal meningitis.The methodology employed is according to approach of the study. The conclusions are consistent with results obtained. The references are appropriate.
The quality of figure 1 must be improved.
The type of letter is not the same in figure 1.
Review the italic style for genera and species of each microorganism and gene.

Comments on the Quality of English Language

None

Author Response

Comments and Suggestions for Authors

The research addressed the main question that the WGS led characterize AMR genes and mobile genetic elements in isolates obtained from fecal samples. The genomic surveillance of AMR enteric bacteria with impact in health public for the design and implementation of AMR interventions and control measures are original and relevant for the field.  Antimicrobial resistance genes are responsible for colistin resistance in E. coli ST219, virulence genes, and plasmid replicons of the Col.  cusC gene in K. quasivariicola ST3879, which induces neonatal meningitis.The methodology employed is according to approach of the study. The conclusions are consistent with results obtained. The references are appropriate.
The quality of figure 1 must be improved.

Response: We have uploaded a PNG format of Figure 1, which has a higher resolution as indicated in Line 236.

The type of letter is not the same in figure 1.

Response: We thank the reviewer for their observation. Although not sure what this means, all letters in Figure 1 have been checked to have the same font and size as indicated in Line 236. Also a footnote with the bacteria specie and corresponding ID have been included in Line 241 –242.

Review the italic style for genera and species of each microorganism and gene.

Response: The italics style for all genera and species across the manuscript have been reviewed and presented appropriately.
